# A Pólya urn approach to information filtering in complex networks

Riccardo Marcaccioli[1] & Giacomo Livan[1,2]

The increasing availability of data demands for techniques to filter information in large complex networks of interactions. A number of approaches have been proposed to extract network backbones by assessing the statistical significance of links against null hypotheses of random interaction. Yet, it is well known that the growth of most real-world networks is non-random, as past interactions between nodes typically increase the likelihood of further interaction. Here, we propose a filtering methodology inspired by the Pólya urn, a combinatorial model driven by a self-reinforcement mechanism, which relies on a family of null hypotheses that can be calibrated to assess which links are statistically significant with respect to a given network's own heterogeneity. We provide a full characterization of the filter, and show that it selects links based on a non-trivial interplay between their local importance and the importance of the nodes they belong to.

[1] Department of Computer Science, University College London, 66-72 Gower Street, London WC1E 6EA, UK. [2] Systemic Risk Centre, London School of Economics and Political Sciences, Houghton Street, London WC2A 2AE, UK. Correspondence and requests for materials should be addressed to G.L. (email: g.livan@ucl.ac.uk)

A vast number of complex interacting systems can be represented as networks[1]. Over the last 20 years, Network Science has been successfully applied in a wide range of disciplines, from Biology to Finance and the Social Sciences[2–6]. One of the main reasons behind such a success is that oftentimes network representations of seemingly very diverse systems share a number of common characteristics. A recurrent feature of several natural and social networks is the lack of a typical scale[2,7], i.e. the marked heterogeneity of major structural features such as the degree or strength distributions.

Understanding which nodes and links represent a set of structurally relevant interactions can be of crucial importance to obtain parsimonious descriptions of complex networks, and, indeed, has contributed to shed light on the functioning of a variety of systems, ranging from biological[8,9], social[10,11], financial[12] or even literature-related[13,14] systems. Furthermore, the size and, in some cases, the density of several real-world networks often prevent any meaningful visualization, and represent a major obstacle for clustering algorithms, which typically work well only with sparse systems[15,16]. Because of such challenges, a number of approaches to extract relevant information from complex networks have been developed over the years. Naturally, any filtering technique hinges on a definition of what type of information represents a signal as opposed to noise. As a result, the network backbones obtained through different filtering techniques carry different meanings and highlight different properties.

Early approaches to filtering focused on proximity networks, and relied on retaining interactions fulfilling some topological constraints. A seminal example of this kind of approach is the minimum spanning tree[17], which selects the tree with the highest total strength embedded in a network. Less constrained generalizations of such method are the planar maximally filtered graphs[18] and the triangulated maximally filtered graphs[19], which reduce topological complexity by forcing the embedding of network backbones on a surface.

Most of the methodologies initially proposed to filter information in weighted networks largely relied on discarding all links whose weights are below a certain global threshold[20–23], leading to backbones not reflecting the multiscale nature of the underlying network[24]. This issue has been addressed by a different class of techniques, which resort to hypothesis testing in order to assess the statistical significance of each link in a network. The disparity filter[25], which arguably represents one of most widely used filtering techniques, falls under this category, and relies on a null hypothesis of uniform distribution of a node's strength over its links. Such a method has been adopted as one of the main benchmarks against which the efficiency of filtering techniques has been tested[26–29].

More recently, a procedure based on a null hypothesis of random connectivity (encoded as the urn problem described by the hypergeometric distribution) has been put forward[30–32]. Other recently proposed methodologies rely instead on frameworks inspired by Statistical Physics, where the properties of empirical networks are tested against those observed in an ensemble of null network models constrained to preserve, on average, the original networks' degree and strength sequences[33,34].

The above procedures provide top-down approaches based on well defined null hypotheses, against which all links in a network are tested individually. While this certainly presents advantages in terms of convenience, at the same time it can lead to a lack of flexibility, as different networks may display different levels of heterogeneity, to which a 'one-fits-all' null hypothesis cannot adapt. Furthermore, most of the above filters are based on null hypotheses of partially random interactions. Yet, interactions in most natural and social systems are far from being random, as past activity naturally breeds further activity[35,36].

Here, we propose a filtering methodology based on a null hypothesis designed to respond to the specific heterogeneity of a network. We shall do so through a statistical test based on the Pólya urn, a well known combinatorial problem driven by a self-reinforcement mechanism according to which the observation of a certain event increases the probability of further observing it. Such a mechanism is governed by a single parameter $a$, which allows to tune the null hypothesis' tolerance to heterogeneity, and to study a continuous family of network backbones $\mathcal{P}_a$. In the following, we shall detail how the Pólya filter works, both from an intuitive standpoint and by providing a full analytical characterization of the family of backbones it generates. In doing so, we shall show how the disparity filter can be recovered, with very good approximation, as a special case of the Pólya filter for $a = 1$. We shall complement our analyses with two case studies to illustrate possible application of the Pólya filter to real-world network data.

## Results

**The Pólya filter**. In the classic Pólya urn problem, we are given an urn containing $B_0$ black balls and $R_0$ red balls. We randomly draw a ball from the urn, we observe its colour and put it back in the urn together with $a$ new balls of the same colour. When this process is repeated $n$ times, the probability of observing $x$ red balls follows the Beta-Binomial distribution[37] with probability mass function $\mathbb{P}(x|n, \alpha, \beta) = \binom{n}{x} B(x + \alpha, n - x + \beta)/B(\alpha, \beta)$, where $B$ denotes the beta function and $\alpha = R_0/a$, $\beta = B_0/a$. In the following, we shall adapt this situation to a network setting.

Let us denote the $N \times N$ symmetric adjacency matrix of an undirected weighted network with $N$ nodes as $W$. An entry $w_{ij} \in \mathbb{N}$ of such a matrix is the weight associated with the link connecting nodes $i$ and $j$, and $w_{ij} = w_{ji} = 0$ when there is no connection between $i$ and $j$. The degree $k_i = \sum_{j=1}^{N} \mathbf{1}(\mathbf{w_{ij}})$ (where $\mathbf{1}$ denotes the indicator function) quantifies the number of connections between a node $i$ and other nodes in the network, while $s_i = \sum_{j=1}^{N} w_{ij}$ denotes the strength of a node $i$, which is a measure of its activity in the network.

With the above notation, we can now rewrite the Pólya urn problem in network terms. Assume we are interested in assessing the statistical significance of a certain weight $w$ falling on one of the links of a node with degree $k$ and total strength $s$. Following the above example, we can think of this as a drawing process from a Pólya urn with 1 red ball and $k - 1$ black balls initially, where we want to measure the probability of drawing $w$ red balls in $s$ attempts. Such a probability reads

$$\mathbb{P}(w|k, s, a) = \binom{s}{w} \frac{B\left(\frac{1}{a} + w, \frac{k-1}{a} + s - w\right)}{B\left(\frac{1}{a}, \frac{k-1}{a}\right)}. \quad (1)$$

The above equation fully describes our class of null hypotheses. We shall assume that a node distributes the weights on its links following a Pólya process whose reinforcement mechanism is governed by the parameter $a$. The rationale of such assumption lays in the flexibility introduced by such a parameter, which naturally captures situations where the more two nodes have interacted, the more further interactions between them become likely. In Fig. 1 we provide a sketch of the Pólya process adapted to a network setting.

Eq. (1) allows to assign a $p$-value to a link of weight $w$ as the sum over all possible 'favourable' outcomes such that at least $w$ red balls have been drawn from the Pólya urn after $s$ draws.

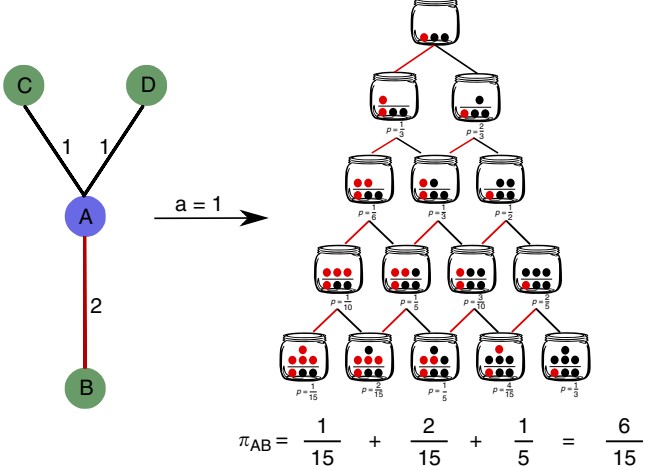

**Fig. 1** Sketch of the Pólya urn process in a network setting. In the toy example shown here, we aim to assess the statistical significance of the link of weight $w = 2$ (highlighted in red) connecting nodes $A$ and $B$, and we wish to do so from the viewpoint of node $A$, whose degree is $k = 3$ and whose strength is $s = 4$. In the Pólya urn analogy, this amounts to starting the urn with one red ball and $k - 1 = 2$ black balls, and computing the probability of drawing at least $w$ red balls in $s$ draws (i.e. the probability that a node distributing its strength $s$ at random through a Pólya process will assign a weight equal or larger than $w$ on the link under consideration). The right part of the Figure shows the possible configurations of the corresponding Pólya urn (for $a = 1$, which entails adding to the urn one ball of the same colour of the latest ball drawn) over the $s$ draws, and their corresponding probabilities computed via Eq. (1). The $p$-value associated to the link is shown at the bottom of the Figure, and is computed as the sum over 'favourable' outcomes (see Eq. (2)), i.e. urns containing at least $w = 2$ red balls (in addition to the one initially present in the urn) at the end of the process

This reads

$$\pi_P(w|k,s,a) = 1 - \sum_{x=0}^{w-1} \mathbb{P}(x|k,s,a), \qquad (2)$$

and in Supplementary Note 1 we provide an explicit formula for this quantity. Once the value of the free parameter $a$ has been set, two $p$-values can be assigned to the weight of each link in the network by applying Eq. (2) from the viewpoint of the two nodes it connects. The statistical significance of a weight is then assessed by comparing its associated $p$-values with a significance level. Since such a procedure involves testing all links in a network, it requires setting a univariate significance level $\alpha_u$ and applying a multiple hypothesis test correction. The two main options available in this respect are the Bonferroni[38] and the false discovery rate (FDR)[39] corrections. The benefits and limitations of the two methods have been largely debated[40,41], and choosing between them essentially boils down to the type of statistical error one is more inclined to accept. The Bonferroni correction is much stricter than the FDR and typically ensures very high precision, leading to a low probability of accepting false positives, at the cost of a potentially low accuracy, i.e. of rejecting true positives. Following[30], in this work we shall adopt the Bonferroni correction: a link of weight $w$ will be validated and included in the Pólya network backbone whenever at least one of its corresponding $p$-values will be such that $\pi_P < \alpha_u/L$, where $L$ is the number of statistical tests performed, which in the case of undirected network is given by twice the number of links in the network (in the case of a link between a node with degree $k = 1$

and a node with $k > 1$ we keep the link only if $\pi_P < \alpha_u/L$ for the node with degree greater than one.).

We have introduced the Pólya filter for weighted undirected networks but it can be easily extended to weighted directed networks (see Supplementary Note 2). In fact, the empirical analyses performed in the following are done on directed networks.

**The backbone family**. As mentioned above, the Pólya filter generates a continuous family of network backbones $\mathcal{P}_a$, which we now seek to characterize as a function of the parameter $a$.

When $a = 0$, the Beta-Binomial distribution (Eq. (1)) reduces to the Binomial distribution with parameters $s$ and $1/k$, i.e. $\mathbb{P}(w|k,s,a=0) = \binom{s}{w}\left(\frac{1}{k}\right)^w\left(1-\frac{1}{k}\right)^{s-w}$. In the urn analogy, the $p$-value associated with a weight $w$ in this case corresponds to the probability of drawing at least $w$ red balls in $s$ attempts with simple replacement from an urn containing 1 red balls and $k - 1$ black balls.

When $a \to \infty$, instead, the Pólya filter loses its dependency on the node strength $s$ and on the weight $w$. This corresponds to a situation where $a \gg k$ balls of the same colour of the first drawn ball are added to the urn, and, as a result, all following draws produce balls of the same colour. Therefore, the probability of drawing at least $w$ red balls is the same of drawing one in the first draw, i.e. $1/k$. This, in turn, leads to an empty network backbone, as the Bonferroni correction criterion cannot be met with such a probability.

Between the two above limit cases, Pólya network backbones monotonically shrink when the parameter $a$ is increased while keeping the statistical significance fixed, i.e.

$$w \in \mathcal{P}_{a_2} \quad \Rightarrow \quad w \in \mathcal{P}_{a_1} \quad \text{for } a_1 \le a_2. \qquad (3)$$

In other words, the largest Pólya set is the one corresponding to $a = 0$, and increasing $a$ progressively removes links from this set. This process is largely driven by a soft dependence of the Pólya filter on the following ratio:

$$r = \frac{w}{s}k = \frac{w}{\langle w \rangle}, \qquad (4)$$

where $\langle w \rangle = s/k$ is the average weight on the links of the node to which the link under analysis is attached. For any fixed value of the parameter $a$, the Pólya filter tends to validate links associated with higher values of $r$. Moreover, higher values of $a$ lead to the progressive rejection of links with higher values of $r$, which in turn leads to the property in Eq. (3). These results are illustrated in Fig. 2 on two network datasets (the 2017 US Airports network and the World Input-Output Database[42], see Methods for a brief description). Indeed, in the two bottom panels one can see that higher values of $r$ tend to be associated with a higher statistical significance (and that such significance, in turn, decreases as $a$ increases), although this is not a strict relationship and there are substantial exceptions. We show in Supplementary Note 4 that these exceptions ensure that thresholding on $r$ does not give a backbone as topologically rich as the one obtained with the full Pólya filter, and therefore the latter should be preferred. This dependence on $r$ is fully described in the Methods section (see Eq. (7)), and is derived analytically in Supplementary Note 3.

In summary, the two quantities that drive the backbone extraction process are $a$ and $r$. First, the ratio $r$ couples a network's local topology (through the degree $k$) to the activity of nodes (through the strength $s$ and weight $w$) in a non-trivial way. The soft dependence of the Pólya filter on such quantity is what ensures that its backbones retain the multiscale nature of the

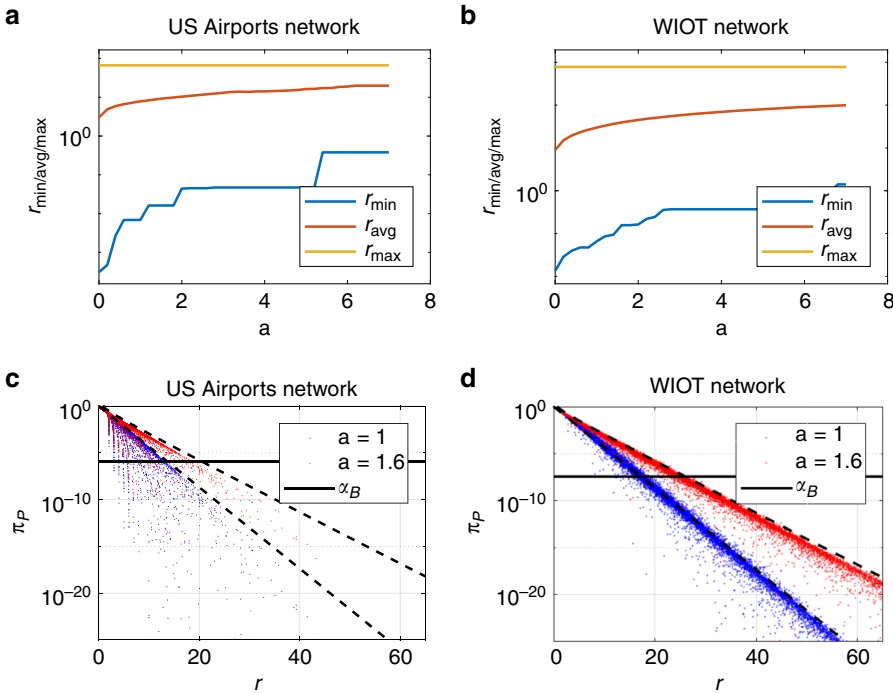

**Fig. 2** Role of the parameter $r$ in the Pólya backbone extraction process. **a** Evolution of the minimum, maximum, and average value of $r$ computed in Pólya backbones for increasing values of $a$ with a univariate significance level $\alpha_U = 0.05$ in the US Airports network. **b** Same quantities computed in the WIOT network. **c** Scatter plots of the $p$-values associated with each link in the US Airports network against the corresponding value of the ratio $r$ for two different values of $a$ at a univariate significance level $\alpha_U = 0.05$. High values of $r$ are associated with $p$-values below the Bonferroni threshold $\alpha_B$ (solid black line), while the opposite is not always true. The black dashed lines illustrate the soft dependence on $r$ described by Eq. (7). **d** Same plot for the WIOT network

networks they are extracted from. The parameter $a$, instead, ensures the flexibility of the method thanks to the analytical control we have over it (see Supplementary Note 5), which can be exploited to tailor the backbone extraction process with respect to the network's own heterogeneity or other meaningful criteria. This will be showcased in the following Section. Moreover, let us mention that $a$ can be directly related to the statistical significance $\alpha$ used to assess the null hypothesis: the backbones generated by taking $a = a_1$ can approximately be considered equivalent to those associated with $a = a_2 > a_1$, provided that a higher statistical significance is set. This is discussed in the Methods Section and numerical evidence for this is provided in Supplementary Note 6.

**Fixing the free parameter**. The main benefit of the Pólya filter is its flexibility, which allows to explore the network backbones obtained when setting different levels of tolerance to heterogeneity, as quantified by the parameter $a$. We devote this section to recommending possible criteria that would identify an optimal value of such a parameter. Clearly, the notion of optimality strongly depends on the specific application being considered. Therefore, we will recommend three different criteria.

- Sweeping: The Pólya filter's monotonicity can be exploited to fix a desired level of sparsity of the resulting backbone with respect to the original network, and to identify the value of $a$ that achieves it. Namely, as a consequence of the property in Eq. (3), the fraction of nodes, of edges, and of total strength retained in the Pólya backbones are all monotonically non-increasing functions of the parameter $a$. Hence, starting from $a = 0$, one can scan the backbone family $\mathcal{P}_a$ for increasing values of $a$ until a desired level of sparsity has been reached (e.g. 5% of the nodes in the original network).

- Maximum likelihood: Eq. (1) can be used to define a log-likelihood function, which can in turn be shown to have a maximum (see Supplementary Note 5). By definition, such a value corresponds to the Pólya process whose self-reinforcement mechanism is the most likely to generate the network under study. Effectively, this amounts to identifying the value $a_{ML}$ corresponding to the 'nullest' model in the Pólya family or, in other words, the Pólya process that best captures the heterogeneity of the network under consideration. We further convey this point in Supplementary Figure 3 by showing on synthetic networks that the maximum likelihood estimates of the parameter $a$ are indeed sensitive to changes in the network's heterogeneity. As such, this criterion is particularly suited to applications where validating the backbone as a whole is a priority. As an example, we report here the values of $a_{ML}$ of the two networks we study in this paper. We find $a_{ML} = 4.5$ for the US Airports network and $a_{ML} = 3.4$ for the WIOT network.

- Salience: Lastly, we are going to propose an ad-hoc criterion based on a compromise between the information retained in a backbone and the information lost by filtering the network it is extracted from. We shall quantify the former in terms of salience[43], a recently proposed yet well established measure of link importance, which can be loosely defined as the fraction of weighted shortest-path trees a link participates in. This is a non-local measure that has been shown to account for both the topological position of a link and for the magnitude of its associated weight (somewhat in analogy to the quantity in Eq. (4)), and captures several essential transport properties. In Supplementary Note 7 we show that, as $a$ increases, the links removed from Pólya backbones are generally those with a lower salience. As a result, the average salience $\langle S(a) \rangle$ retained in the backbones $\mathcal{P}_a$ increases with $a$.

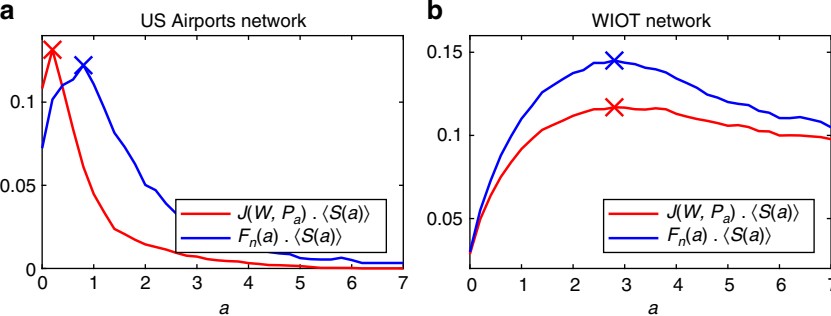

**Fig. 3** Optimality measures $O_1$ and $O_2$. These are calculated on the extracted backbones (at a univariate significance level $\alpha_u = 0.05$) as a function of $a$. The optimal values are highlighted with a cross. **a** Optimality measures for the US Airports network. The optimal values are $a^* = 0.2$ for $O_1$ and $a^* = 0.8$ for $O_2$, respectively. **b** Same plot for the WIOT network. The optimal values are $a^* = 2.8$ for both $O_1$ and $O_2$

Measuring the quality of a backbone just in terms of average salience could lead, in most cases, to an excessive depletion of the network under study. This tendency can be contrasted by penalizing large differences between backbones and their original networks. We do so by introducing the two following optimality measures

$$O_1 = J(W, \mathcal{P}_a) \cdot \langle S(a) \rangle, \qquad O_2 = F_n(a) \cdot \langle S(a) \rangle, \qquad (5)$$

where we are weighting the average salience against the Jaccard similarity $J(W, \mathcal{P}_a)$ between the weights in the original network and those in the backbone, or against the fraction $F_n(a)$ of nodes retained in $\mathcal{P}_a$, respectively. Figure 3 shows the behaviour of the above metrics as functions of $a$ in the two networks we study. As it can be seen, both metrics achieve a maximum $a^*$, which represents the optimal compromise between high salience and similarity with respect to the original network.

**Comparisons with other network filters.** In this Section and in Supplementary Note 8 we further characterize the Pólya filter's family of backbones through the comparison with some of the other available filtering techniques. In a nutshell, this will allow us to show that Pólya backbones are typically sparse, salient and heterogeneous.

Figure 4 shows different properties of the Pólya backbones of the US Airports and WIOT networks obtained for different multivariate significance levels $\alpha$ with those of the backbones obtained at the same statistical significance with the Hypergeometric Filter (HF)[30], the Maximum-Likelihood filter (MLF)[33], the Enhanced Configuration Model (ECM) based on the canonical ensemble constrained both on degrees and strengths[34], the Noise-Corrected (NC) Bayesian filter proposed in[44], and the Disparity Filter (DF)[25], which in the Methods Section and in Supplementary Note 3 we show to correspond to a large-strength approximation of the Pólya filter for $a = 1$. Comparisons with the GloSS filter[28] were also performed, but their results are not reported due to the excessive sparsity of the backbones produced by such method when accounting for multiple hypothesis testing.

As it can be seen from the two upper panels (see also Supplementary Note 8), Pólya backbones are considerably more parsimonious than those provided by the other filters considered. This is especially true when correcting for multiple hypothesis testing (the black vertical lines in each plot correspond to a Bonferroni-corrected univariate significance level of 0.05, which is crucial to reduce the number of false positives retained in the backbones. In addition, when setting $a \simeq a_{ML}$ (see previous Section), the Pólya filter generates ultra-sparse backbones whose

links are statistically significant with respect to the network's own heterogeneity. This will be further illustrated with a case study in the following Section.

The two middle panels show values of the optimality measure $O_1$ as a function of statistical significance. As it can be seen, for a wide range of the parameter $a$ the Pólya filter is able to strike a good balance between sparsity and salience, a property that is not shared by any other of the methods considered.

The two bottom panels demonstrate the heterogeneity of Pólya backbones, by showing the Jaccard similarity between the $B$ weights retained in a backbone and the top $B$ weights in the original network. This essentially amounts to assessing how heterogeneous a network backbone is with respect to a 'naive' backbone obtained simply by thresholding on weights. As one can see, the Pólya filter generates backbones that are considerably more heterogeneous than those provided by the other methods, with the exception of the NC filter when applied to the WIOT network, where, however, such filter ends up discarding the more salient links.

The two bottom panels also show that the Pólya filter is more responsive to statistical significance than the other methods. Indeed, Pólya backbones are built around complex and sparse cores that correspond to links associated with very low $p$-values. As the threshold $\alpha$ increases, such cores are enriched by links with heavier weights which are structurally important for the network but classified as less statistically significant. Diversely, the other methods are much less responsive to $\alpha$, even when varied across several orders of magnitude.

The above properties are inherited by the disparity filter, which, as demonstrated in the Methods Section and in Supplementary Note 3, is a large-strength approximation of the Pólya filter for $a = 1$. In most cases (see also those in Supplementary Note 8), the disparity filter generates rather parsimonious backbones that are more salient and heterogeneous than most of the backbones produced by the other methods considered above. Yet, depending on the specific application or network, the disparity filter might be far from optimal within the Pólya family. This is the case, for example, in the US Airports network, where the disparity filter backbone is rather sub-optimal in terms of salience, as demonstrated by the comparatively low value of $O_1$ it achieves within the Pólya family.

All in all, the above results reiterate that the Pólya filter's main advantage lies in its flexibility, which allows to tune the filter to the specific network or application under consideration. Moreover, the filter's ability to 'compress' the salience and heterogeneity of the original networks in ultra-sparse backbones is unmatched by the other methods we considered. In the next section we show how these properties can be exploited in order to gain insight on real-world networks.

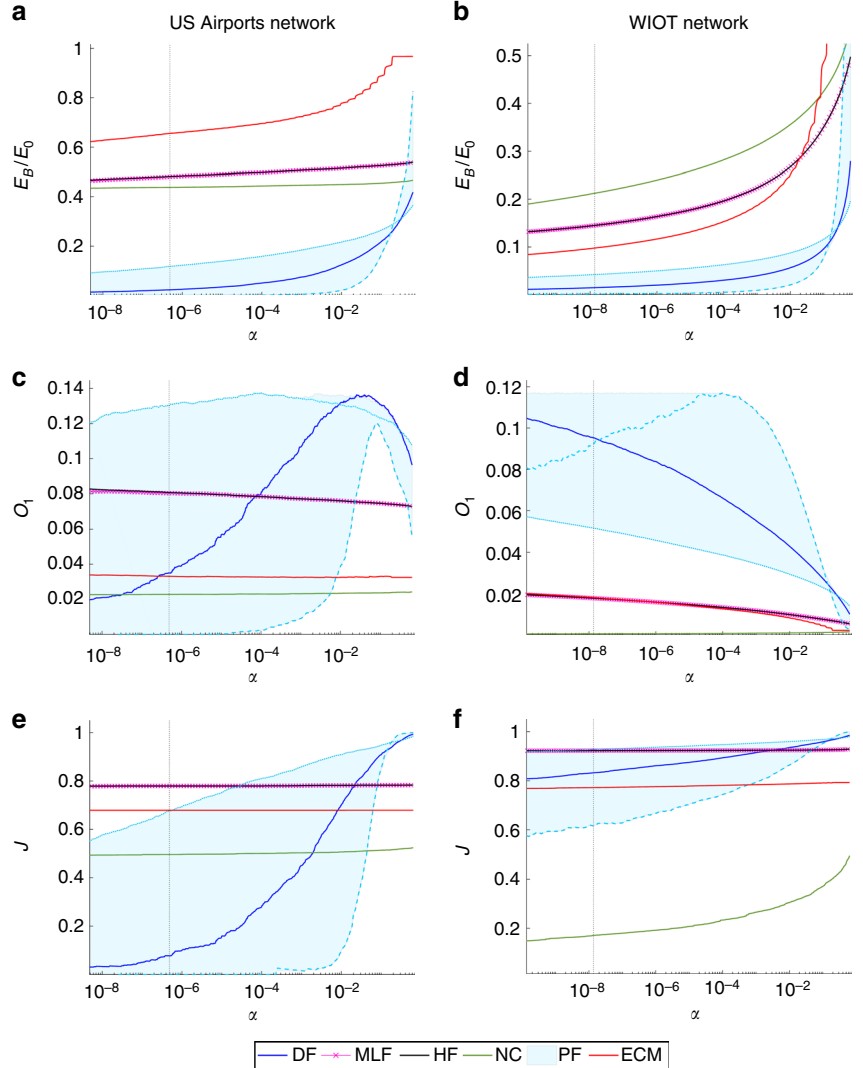

**Fig. 4** Comparisons between the backbones generated by the Pólya filter (PF) and other network filtering methods. The methods we consider are the Hypergeometric filter (HF), the Maximum-Likelihood filter (MLF), the Enhanced Configuration Model (ECM), the Noise-Corrected filter (NC), and the Disparity filer (DF), which corresponds to a large-strength approximation of the the Pólya filter for $a = 1$ (see the main text for references to the papers where those methods were introduced). All quantities are shown as a function of the multivariate significance level used in the tests. **a**, **b** Fraction of links retained in the backbones with respect to the total number of links in the original networks. **c**, **d** Value of the salience-related measure $O_1$ defined in Eq. (5). **e**, **f** Jaccard similarity between the $B$ weights retained in the backbones and the top $B$ weights in the original networks. In all plots the light blue band correspond to all values measured in the Pólya backbone family for $a \in [0.2, 7]$, with the light blue solid (dashed) line corresponding to $a = 0.2$ ($a = 7$); vertical dashed lines correspond to the Bonferroni-corrected 5% significance level

The above observations can be largely replicated based on the additional comparisons shown in Supplementary Note 8 between the above methods and the Pólya filter.

**The short-haul backbone of the US Airports network**. In the following we show how the Pólya filter can be used to gather unique insights on the US Airports network.

Figure 5 shows the Pólya filter's backbones of the US Airports network obtained for different values of the filter's parameter $a$. Thicker lines correspond to 'heavier' links (i.e. routes with more passengers), while lines in blue, orange, and purple correspond, respectively, to short, medium, and long-haul flights according to the US Bureau of Transportation's classification.

As per Eq. (3), higher values of $a$ lead to sparser backbones. The backbone in the top-left panel corresponds to $a = 0.4$ (which is between the two values of $a$ that optimize the metrics defined in

Eq. (5)), and is the most salient one. As such, it features the most crucial long-haul connections between hubs and/or the more geographically remote states (Alaska, Hawaii, and Puerto Rico). Most, although not all, of such connections are retained when setting $a = 1$, which approximately corresponds to the disparity filter's backbone, shown in the top-right panel.

Things change considerably when increasing the filter's tolerance to heterogeneity through higher values of $a$. The backbone in the bottom-left panel is the one obtained for the highest value of $a$ that still allows to retain both connections between New York and Los Angeles ($a = 2.6$), i.e. the two largest American cities. Notably, these are the only two long-haul connections remaining. Finally, when tuning the filter's tolerance to the network's own heterogeneity ($a = a_{ML} = 4.5$), we obtain an ultra-sparse backbone, shown in the bottom right panel, where all long-haul flights and almost all connections between major cities and hubs have been filtered out. In Supplementary Note 9 we

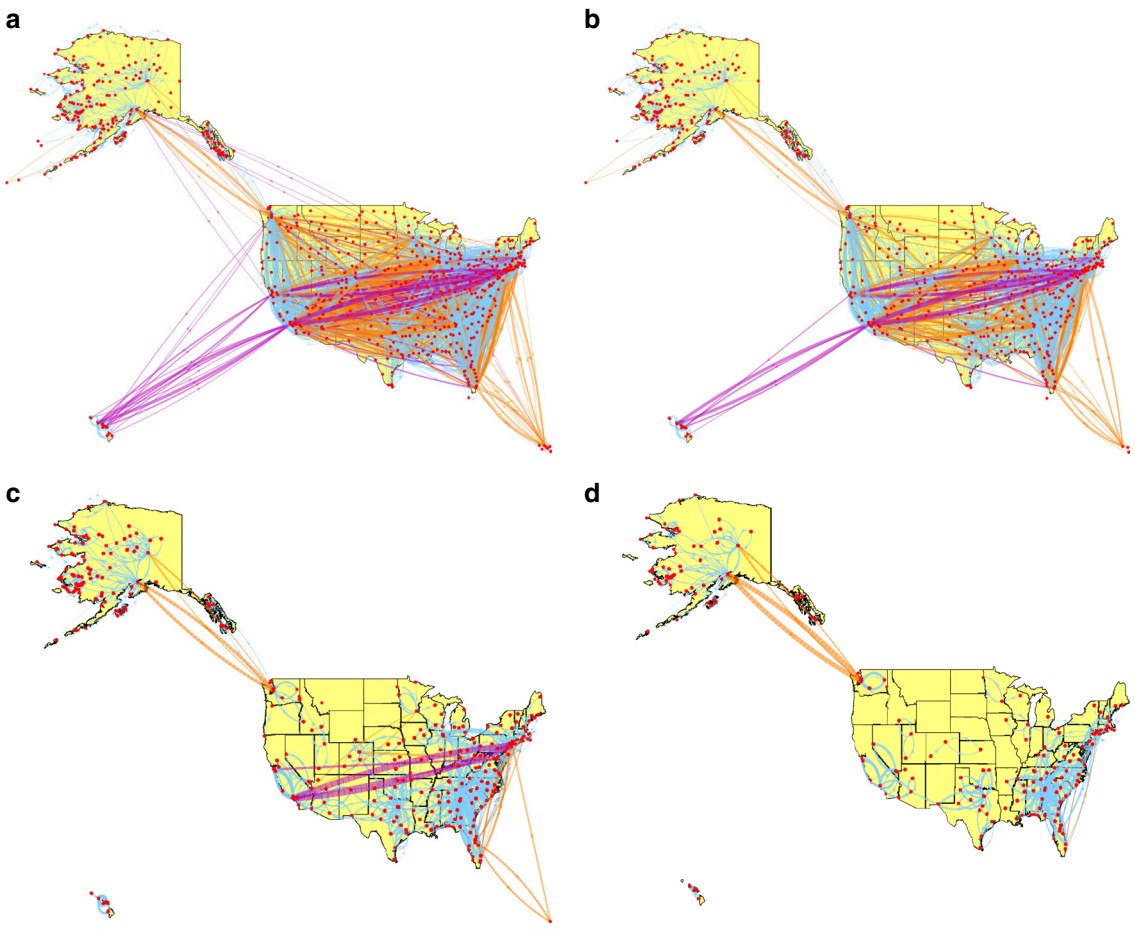

**Fig. 5** Pólya backbones of the US Airports network for different values of the filter's parameter $a$. **a** Backbone for $a = 0.4$ (which is an intermediate value between the two that optimise the salience metrics in Eq. (5)), where most long-haul flights between hubs are retained. **b** Backbone for $a = 1$, approximately corresponding to the one obtained via the disparity filter. **c** Backbone for $a = 2.6$, which is the highest value of the filter's parameter where a long-haul flight (New York–Los Angeles) is retained. **d** Backbone for $a = a_{ML} = 4.5$, where all long-haul flights and all connections between hubs have been filtered out

further characterise such a backbone by showing its projection onto US states, and showing that it is mostly made of two-way links between bordering or geographically close pairs of states. This is because long-haul connections are precisely those that determine the network's heterogeneity, while the links retained are those identified as statistically significant with respect to it. The only major hub still involved in a large number of connections is Atlanta, which is the busiest airport in the world and serves almost 20% more passengers than the second busiest US airport. Notably, the links retained form a network of mostly regional and short-haul flights connecting airports that are often of secondary importance on the national scale. Yet, these flights provide vital connections, carrying very large numbers of passengers relative to the overall heterogeneity of the broader transport system they are embedded in. This is well exemplified by Alaska, where a very large number of internal flights are validated.

**Predicting trade in the WIOT network.** As an example of a practical use of our methodology, we show how the out-of-sample performance of a simple econometric model aimed at predicting trades in the WIOT network can be improved by using the Pólya filter.

Understanding technological innovation ultimately hinges on the ability to foresee structural changes in the relationships between economic actors. Several studies have recently looked at

this issue from a network perspective, where firms purchase goods from each other and combine them into more technologically sophisticated products (see, e.g.[45]). Within this framework, being able to predict changes in trading relationships can be of crucial importance in order to anticipate technological shifts and allow for an efficient allocation of investments.

Here, we follow[45,46] and build a simple model to predict trading relationships in the WIOT dataset based on its network properties. We refer to Supplementary Note 10 for a detailed description of the model. In short, it is a linear regression model aimed at predicting the future trading volume between two industrial sectors based on the relative importance of their past trading volume (with respect to their overall trading volume) and on their proximity in the network computed via the Leontief input–output matrix[47].

We exploited such model to assess the potential benefits gained in terms of prediction accuracy when employing the Pólya filter. Namely, we constructed Pólya backbones of the annual WIOT networks from 2006 to 2010 both for $a = 1$ (which essentially corresponds to the disparity filter) and for $a = a_{ML} = 3.2$. We used such backbones to calibrate the model (see Supplementary Table 1 for the model's coefficients and their significance) and to make out-of-sample predictions of the trading volumes of the links marked as significant in the three following years. We compared the predictive power of such models with that of the model calibrated on the full unfiltered WIOT network.

**Table 1 $R^2$ coefficients of the model calibrated on the three different datasets when used to make out-of-sample predictions**

|  | Out-of-sample $R^2$ | | |
|---|---|---|---|
|  | **2011** | **2012** | **2013** |
| Unfiltered Networks | 0.1349 | 0.1371 | 0.1367 |
| Backbones $\mathcal{P}_{a=1}$ | 0.1960 | 0.1989 | 0.1972 |
| Backbones $\mathcal{P}_{a_{ML}}$ | 0.2242 | 0.2181 | 0.2127 |

In Table 1 we compare the predictive power of the model when calibrated on Pólya backbones and on the full, unfiltered, WIOT network in terms of out-of-sample $R^2$ coefficients. As it can be seen, applying the Pólya filter substantially improves the percentage of variance in the data explained by the model, with the best results being obtained when applying the filter for $a = a_{ML}$.

These results further testify that the information contained in Pólya backbones is substantial. Indeed, the full WIOT network contains $2.68 \times 10^6$ links, whereas the two Pólya backbones employed above contain $4.89 \times 10^4$ and $1.48 \times 10^4$ links for $a = 1$ and $a = a_{ML}$, respectively (see Supplementary Table 1). This, in turn, means that the information lost by reducing the number of links by two orders of magnitude is more than offset by the higher overall informativeness of the networks generated by the filter.

## Discussion

In the era of Big Data, information filtering methods are needed more than ever to handle the dazzling complexity of both social and natural networked systems. In this paper, we have proposed a technique based on the Pólya urn model to extract backbones of statistically relevant interactions between pairs of nodes in a network. In the network context, the parameter $a$ tuning the Pólya model's self-reinforcement mechanism effectively becomes a tolerance to a network's heterogeneity. This, in turn, introduces an element of flexibility, which, to the best of our knowledge, other network filtering techniques do not provide.

Indeed, we have shown that the Pólya filter generates a continuous family of network backbones. Depending on the specific application, the null hypothesis underpinning the filter can be chosen so as to have a different tolerance to heterogeneity. The low-tolerance regime ($a < 1$) corresponds to a rather loose filtering, suited to situations where the main goal is to filter out interactions that can be unquestionably identified as noise. On the other hand, the high-tolerance regime ($a > 1$) corresponds to increasingly restrictive tests, where only links of substantial structural importance survive.

As we have shown, the link selection criterion underpinning the Pólya filter is based on the interplay between topology and the local relative importance of a link, quantified by the parameter $r$. This, in turn, guarantees that the filter does not perform a naive link selection merely based on retaining high strength links connecting hubs, but instead ensures a non-trivial scanning of all the relevant scales of a network.

## Methods

**Data**. In the following we provide a short description the datasets we employed to illustrate the Pólya filter.

World Input-Output Database: The Database contains yearly aggregate economic transactions, measured in millions of dollars, between the industrial sectors of different countries from 2000 to 2014. The database features transactions between 64 sectors in 45 countries[42,48]. The resulting series of networks and their properties have been analysed extensively in a number of studies[49–51]. The dataset

we are going to use in this paper is the 2014 network, which features 2464 nodes and 738,374 edges.

US Airports network: The dataset contains information on the flights between a number of US airports during the year 2017. Each link represents a connection between airports, with the weight representing the number of passengers on all flights on that route in the given direction. The system contains 1151 airports and 20,580 different connections. The same network with data coming from different years has already been used in network filtering literature[25,34].

In Supplementary Note 8 we show comparisons between the Pólya filter and other filtering techniques on the two following additional datasets.

High School network: This dataset reports face-to-face interactions between students recorded in 2013 in a Marseille high school throughout a period of five days[52]. The weights on the network's links correspond to the number of interactions recorded during the experiment, and interactions were recorded every 20 s. The network is made of 5818 weighted interactions among 1567 students.

Florida ecosystem network: Weights in this network represent the carbon exchanges between taxa in the cypress wetlands of South Florida during its dry season[53]. The network is formed of 128 nodes and 2137 links.

**Approximations of the Pólya filter's $p$-values and relationships with the disparity filter**. Eq. (2) can be considerably simplified assuming $s \gg k/a$, and $w \gg 1$. In this regime, the $p$-value the Pólya filter associates to a weight $w$ on a link belonging to a node with degree $k$ and strength $s$ reduces to

$$\pi_P(w|k,s,a) \approx \frac{1}{\Gamma\left[\frac{1}{a}\right]} \left(1 - \frac{w}{s}\right)^{\frac{k-1}{a}} \left(\frac{wk}{sa}\right)^{\frac{1}{a}-1}, \tag{6}$$

where $\Gamma$ is the Gamma function. The rigorous derivation of the above approximation is provided in Supplementary Note 3, where we also show numerically that the approximations used to derive Eq. (6) hold for large fractions of edges. If we further approximate Eq. (6) by expanding it around $w/s \approx 0$ we obtain

$$\pi_P \approx \frac{e^{-\frac{r}{a}} \left(\frac{r}{a}\right)^{\frac{1}{a}-1}}{\Gamma\left[\frac{1}{a}\right]}, \tag{7}$$

where $r$ was introduced in Eq. (4). This result demonstrates the soft dependence of the Pólya filter on the ratio $r$ mentioned in the main text and shown in Fig. 2.

Notably, when Eq. (6) holds, the Pólya filter does not depend on $w$ and $s$ separately (as it normally does, as per Eqs. (1) and (2)), but only depends on such quantities through the ratio $w/s$ and the $p$-value loses its ability to discriminate between nodes with different heterogeneity. As we shall see in the following section, this allows to extend the applicability of the Pólya filter to networks with non-integer weights.

Setting $a = 1$ in Eq. (6) gives $\pi_P = (1 - w/s)^{k-1}$, which coincides with the $p$-value prescribed by the disparity filter[25], i.e.

$$\pi_D(w|k,s) = 1 - (k-1) \int_0^{w/s} (1-x)^{k-2} dx = \left(1 - \frac{w}{s}\right)^{k-1}. \tag{8}$$

We can therefore conclude that the disparity filter corresponds to a large strength approximation of the Pólya filter in a special case ($a = 1$). This is demonstrated in Fig. 6, where we plot the relationship between the $p$-values assigned by the Pólya and disparity filters to the same links. As it can be seen, the two sets of values are indeed very close when $a = 1$. This should not come as a surprise. Indeed, the null hypothesis underlying the disparity filter is ruled by a particular case of the Dirichlet distribution, which is known to be a limit case of the Beta-Binomial distribution as the number of draws goes to infinity[54].

The relationship between the Pólya and disparity filters is further investigated in Supplementary Notes 3 and 6.

**Equivalence of Pólya backbones**. In this Section we are going to show that the backbones produced by the Pólya filter for different values of $a$ can be made approximately equivalent by tuning the filter's statistical significance.

Assessing the statistical significance of a link with weight $w$ (or associated to a value $r$ of the ratio in Eq. (4)) entails determining whether it is compatible with the assumed null hypothesis. Using a Gaussian analogy, we can say that a value $r$ is compatible with the null hypothesis if $\mu_r(a) - b\sigma_r(k, s, a) < r < \mu_r(a) + b\sigma_r(k, s, a)$, where $b \geq 0$ is inversely proportional to the statistical significance $\alpha$, while $\mu_r$ and $\sigma_r$ denote the expected mean and standard deviation of the ratio $r$ under the Pólya null hypothesis. These read:

$$\mu_r(a) = \mathbb{E}[r] = 1 \tag{9}$$

$$\sigma_r^2(k,s,a) = \mathbb{E}\left[(r - \mu_r)^2\right] = \frac{k-1}{s} \frac{k+as}{a+k}. \tag{10}$$

Let us then consider the null hypothesis associated with two different values $a_1$ and $a_2$ of the parameter, such that $a_2 \geq a_1$, and look for a scaling parameter $c$ that

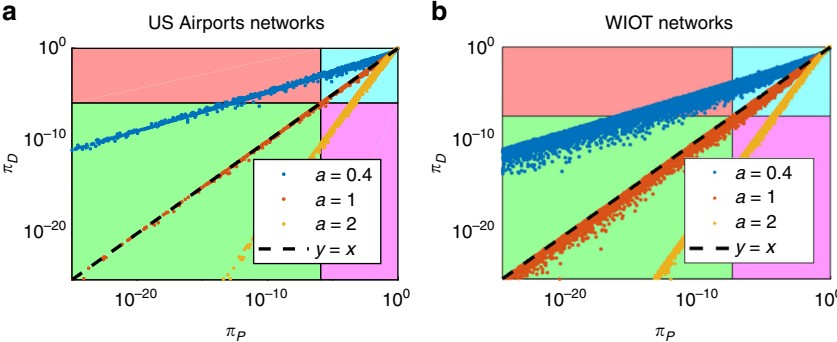

**Fig. 6** Comparison of the $p$-values prescribed by the disparity ($\pi_D$) and Pólya ($\pi_P$) filters computed for different values of $a$ (at a univariate significance level $\alpha_u = 0.05$). Each region of the plot is coloured depending on the significance of the two filters. Points in the blue (green) region correspond to links rejected (accepted) by both filters, while points in the purple (red) region correspond to links accepted only by the disparity (Pólya) filter. **a** $p$-values computed on the US Airports network. **b** $p$-values computed on the US Airports network

makes them equivalent. In order to do so we just need to impose:

$$\mu_r(a_1) \pm \sigma_r(k, s, a_1) = \mu_r(a_2) \pm c\sigma_r(k, s, a_2), \quad (11)$$

for $c \geq 0$. Using $\mu_r(a_1) = \mu_r(a_2) = 1$, and setting $a_2 = da_1$ (with $d \geq 1$), we can solve the above equation for $c$ and get

$$c = \sqrt{\frac{a_1 + k/d}{a_1 + k} \frac{a_1 s + k}{a_1 s + k/d}}, \quad (12)$$

which is a monotonically decreasing function of $d$. This means that the same backbone produced by the Pólya filter for $a = a_1$ can be approximately reproduced with $a = a_2 \geq a_1$ and a smaller region of compatibility with the null hypothesis (i.e. a higher statistical significance). In other words, in the Pólya filter family of backbones, tolerance to heterogeneity and statistical significance are closely related.

**Networks with non-integer weights**. The Pólya filter is encoded in Eq. (1), which depends on $w$ and $s$ individually. This means, that Eq. (1) is able to discriminate between nodes with different heterogeneity (given a fixed value of $k$), e.g. between two nodes characterised by the pairs $(w, s) = (10,100)$ and $(w, s) = (100,1000)$, respectively. This feature is naturally suited to deal with integer weights, such as those coming from counting experiments (e.g. as in the US Airports network).

The above property vanishes when $s \gg k/a$ and $w \gg 1$, leading to Eq. (6), which only depends on the ratio $w/s$ and, in fact, should be exploited to apply the Pólya filter when dealing with networks with non-integer weights, even in cases when such approximations do not hold. Of course, doing so will change the underlying null hypothesis: indeed, Eq. (6) does not assign a $p$-value to a weight $w$, but rather to a rate of interaction $w/s$. In most cases the $p$-values given by Eq. (1) and (6) are practically the same (see Supplementary Fig. 1), and can be used interchangeably when dealing with integer weights. Conversely, Eq. (1) cannot assign $p$-values to non-integer weights, but in such cases one can always assign a $p$-value to the interaction rate $w/s$ through Eq. (6).

We can further justify the use of Eq. (6) by thinking of an overall rescaling of the weights by a large factor $c$. For example, let us consider a network whose lowest weights are of order $10^{-4}$. Applying Eq. (1) to such a network would entail rescaling its weights by a factor $c \geq 10^4$ before filtering. Doing so, however, automatically takes us to the regime under which Eq. (6) holds (i.e. $s \gg k/a$ and $w \gg 1$), which therefore becomes the Pólya filter's analytical expression for non-integer weights.

**Code availability**. The MATLAB code used to implement the Pólya filter in this study is available at http://mathworks.com/matlabcentral/fileexchange/69501-polya-filter.

## Data availability
The US Airports network data used in this study are available at https://www.bts.gov/; the World Input-Output database is available at http://www.wiod.org/home; the Florida ecosystem network is available at http://konect.uni-koblenz.de/; the high school network is available at http://www.sociopatterns.org.

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

## Acknowledgements
G.L. acknowledges support from an EPSRC Early Career Fellowship in Digital Economy (Grant No. EP/N006062/1). We thank R.N. Mantegna for reading the preliminary version of our manuscript.

## Author contributions
R.M. and G.L. designed research; R.M. performed research and analysed data; and R.M. and G.L. wrote the paper.

## Additional information

**Competing interests:** The authors declare no competing interests.

