## [Peer Review File · Nature Communications]

Reviewer #1 (Remarks to the Author):

This paper introduces a novel class of network filtering techniques based on Polya urn models. The manuscript is very technical, but conveys the general idea behind the method proposed. A very nice feature of the method proposed is to generalize the well-known disparity filter. The generalization consists in providing a tolerance parameter that can be used to tune how strict the filter is in the selection of edges to preserve. The paper is well written, and the results of the analysis convincing enough. I am definitely in favor of publication.

Few remarks are in order, and I hope that the authors will implement them in a revised version of the manuscript:

- 1) I believe that a schematic figure representing how the network null model is generated would be helpful.
- 2) I am not entirely sure that some of the mathematical expressions placed in the main text should actually go there. For example, Eq. (2) is a nightmare. What value does the equation bring to a generic reader of a multidisciplinary journal as Nat Comm?
- 3) Is the implementation of the filter publicly available? This would represent a valuable addition to the paper.
- 4) The choice of the colors in the figures is not optimal. They are fine in the color version, but I am afraid they will appear indistinguishable in a B&W version.
- 5) Don't use vectorial files for this type of figures. They contain too many points, and make the pdf file unmanageable!

Reviewer #2

This paper presents a new method to filter links of weighted networks. The aim is to keep the links that carry the relevant structural information of networks. The authors argue that the state-of-the-art method Disparity Filter [1] (DF) is not good enough, because the null hypothesis it applies is too rigid and it tends to keep most of the links associated with large weights. A new method, the Pólya Filter (PF), is proposed in this paper to address these issues. The authors consider a more non-random null hypothesis by incorporating the self-reinforcement mechanism into the generation of link weights. A parameter a is used to control the strength of this effect and the freedom in selecting a adds flexibility to the new PF method. It is expected that, the incorporation of the self-reinforcement mechanism will allow more high-weighted links to be compatible with the null hypothesis and the PF will therefore keep fewer high-weighted links and obtain more reasonable results.

Three criteria are provided for choosing a and the second one is interesting, which is to fit the model used in the null hypothesis to the network and obtain the MLE of a . This procedure certainly makes the new method more adaptive to networks' heterogeneity.

The authors further justify the superiority of the new method by demonstrating that, for each link (between nodes i and j), PF's decision is made approximately referring to the magnitude of $r_{i,j} = w_{i,j}k_i/s_i$ and $r_{j,i}$, where k_i and $s_i = \sum_j w_{i,j}$ are respectively the number of links and the strength of node i . The authors argue that r couples the local topology (through k) with the activity of the nodes (through s and w) and provides a good indication of whether a link should be kept or not.

The idea behind PF of adapting the Pólya urn model to the generation of network link weights is novel and it does successfully incorporate the consideration of self-reinforcement into the null hypothesis. However, real data examples and more comparisons/analyses are needed to justify that the new method is necessary and that it can achieve what the previous method (DF) can not do.

Major Concerns

- The authors did not provide real data examples to demonstrate that the previous method DF will fail because it does not take into account networks' own heterogeneity. In order to validate the significance of the new method, real data experiments are better included to compare the results of DF and PF. Besides, the authors' concerns on the over-selection and inflexibility issues of DF are better echoed by real data examples.
- The authors argue that the deficiency of DF is partly due to its inclination to select most high-weighted links and its rigidity in the null hypothesis. However, a simple way to address these problems is to adjust the significance level α used in hypothesis testing. For example, the over-selection problem can be solved by decreasing α . In order to make more informative comparisons, the authors should compare PF with DF (with adjustable α 's) and identify the effects that can be achieved by PF only.
- In the section "Heterogeneity and weight rescaling", the authors discuss how the procedure of rescaling all weights with $c \in \mathbb{R}$ affects PF. They demonstrate that the set of selected links vary with respect to c . This property seems to be a disadvantage of the new method, because in real data sets, the absolute scale of link weights could be rather arbitrary. It may not be sensible to let this scale to affect the result of filtering. In contrast, the result of DF is not affected by this scale.

- The authors argue that one merit of PF is its dependence on r . However, they should notice that DF also has this property. Therefore, this dependence does not necessarily help to show that the new method is better than previous ones.

Minor Concerns

- The section “The backbone family” discusses that by varying a , PF is able to obtain a monotone family of link sets. Similar results can also be achieved by DF with varying α 's. A question similar to that in the second bullet point of the major concerns is, what are the merits of the family obtained by PF compared to that obtained by DF.
- The authors discuss that the soft dependence of PF on $r = wk/s$ ensures the multiscale nature of the network, while links with small r 's still have a chance to be validated by PF. One question arising from this argument is: how is PF better than the method of doing direct thresholding on r , which seems to be both reasonable and quite simple. More discussions and justifications of PF are needed here.
- The PF is presented partly to address DF's over-selection problem of high-weighted links. A natural concern is, will the new method suffer from the problem of under-selection, especially when a is selected by the MLE procedure. Discussions and numerical experiments are better provided to address this concern.
- In the second paragraph of the section “The role of the r ratio”, the authors discuss how the outcome of PF varies, when w/s is fixed and k varies. It seems to me that this statement is problematic, since usually when k increases, w/s will decrease. It is natural to decouple the variables s/k or w from k but not w/s from k .
- The calculation of the p -value of the new method involves the generalized hypergeometric function. As far as I know, the result of a generalized hypergeometric function is not always easy to obtain and I'd like to see more comments about this computational issue here.
- The idea that the section “Heterogeneity and weight scaling” tries to convey is a little tangled and the purpose of this section is not clear.

More Suggestion

- The authors viewed r as a combination of network's local topology (k) and the activity of nodes (w and s). Nevertheless, a simpler way to view r is $r_{i,j} = w_{i,j}/\bar{w}_i$, where \bar{w}_i is the average weight of the links of node i . With this formulation, $r_{i,j}$ is simply the standardized weight of the i - j link. When the effect of PF is analyzed, this way of understanding of r might be helpful.

References

- [1] Serrano, M. Á., Boguná, M., and Vespignani, A. (2009) Extracting the multiscale backbone of complex weighted networks, *Proceedings of the National Academy of Sciences*, 106(16), 6483-6488.

Reviewer #3 (Remarks to the Author):

The authors proposed a method to extract network backbones by assessing the statistical significance of links. Inspired by the classical Polya urn model, they applied the model to network and assumed that a node distributes the weights on its links following a Polya process. The statistical significance of a link with weight w can be quantified by its two p -values that obtained from the viewpoint of its two endpoints. Generally speaking, the method is somewhat innovative. The paper is well organized but not well written for general readers. Consider that Nat. Comm. aims at broad readership and expects significant scientific advance, it could be better to be published in a more specialized journal. Here are my specific comments:

1. The model only considers the local information of the nodes, how to reflect the influence of network global structure on the significance of links.
2. In the introduction, the authors reviewed many approaches to filtering, which were not used for comparison in the later experiments. Comparisons should be added. Besides the mentioned methods, there are many network-based approaches to find the network skeleton.
3. In equation (4), s and k are for original network or for Polya network? The indicator r is for node or for link? If the latter, which endpoint will be used in equation (4), or there are two r from the viewpoint of the two endpoints. What is the mathematical relation between r and a in equation (4)?
4. If the equation(6) is more suitable for the network with uniform degree distribution and heterogeneous weight distribution? Heterogeneous degree distribution may lead to a larger error. Therefore, I'm wondering if Polya filter is a general model that can be applied to all kinds of networks. This should be discussed to show in which condition the model performs well.
5. Why the performances on the two networks shown in figure 2 are much different? How do network structural features matter?
6. How to evaluate the performance of Polya filter with other benchmarks? One possible method is network dismantling. If a network collapses faster after removing the links according to their significance given by Polya filter than other methods, this to some extent shows the superiority of this method. Other evaluation methods can also be applied.
7. The method should be tested on more disparate networks to show its performance. The results can be put in SI.
8. Is this method only for the weighted network? Can it be also applied to unweighted networks where w values for all links are 1 leading to $r=1$.
9. How to apply Polya filter on the networks with non-integer weights, e.g. $w=0.1, 2.5...$
10. The paper should be rewritten to improve the readability to make it be of interest to the broad readership of Nat. Comm.. For instance, one can add one simple example network (with 6-8 nodes) to illustrate the model and make a comparison with other 2 or 3 benchmarks. Besides, it will be much better if the authors provide some real applications of their method.

Response to the reviews of submission NCOMMS-18-25831 “A parametric approach to information filtering in complex networks: The Pólya filter”

We thank the Reviewers for their comments and feedback, which gave us the opportunity to substantially improve our paper. We have extensively rewritten it in order to make it more suitable to a multidisciplinary scientific audience such as Nature Communications’ readership. Moreover, following the Reviewers’ suggestions, we added two new Sections to the paper in order to (i) show comparisons between the Pólya filter and five well-established network filtering techniques, and (ii) to show concrete applications of the Pólya filter to network data. In addition, based on the feedback from Reviewer #2, we have further characterised the relationship between the Pólya and disparity filters, both analytically and numerically.

In the following, we reply to all comments from the Reviewers in the same order as they appear in their reports.

Reviewer #1

I believe that a schematic figure representing how the network null model is generated would be helpful.

We thank the Reviewer for suggesting this. We have added a schematic representation of the Pólya process associated with a simple network toy example. This is now Figure 1 of the revised manuscript. We hope this will contribute to improve the paper’s readability and will make the ensuing technical points easier to understand.

I am not entirely sure that some of the mathematical expressions placed in the main text should actually go there. For example, Eq. (2) is a nightmare. What value does the equation bring to a generic reader of a multidisciplinary journal as Nat Comm?

Following the Reviewer’s suggestion, we have removed Eq. (2) in its previous form and moved it to Supplementary Note 1. We have kept the sum over the probabilities of Eq. (1), as that is the expression we use to compute p -values numerically (see also our reply to a related point raised by Reviewer #2). As mentioned above, we have moved the “heaviest” mathematical expressions and technicalities to the Methods Section and/or the Supplementary Information document.

Is the implementation of the filter publicly available? This would represent a valuable addition to the paper.

We thank the Reviewer for suggesting this possibility, which we had not thought about. We have now posted online a commented Matlab code at mathworks.com/matlabcentral/fileexchange/69501-polya-filter, and referenced it in the paper.

The choice of the colors in the figures is not optimal. They are fine in the color version, but I am afraid they will appear indistinguishable in a B&W version.

We are not sure which figures the Reviewer is referring to. We suppose it was the figure with the two network visualizations. Due to feedback from the other two Reviewers, we have now removed it and replaced it with other plots and visualizations, which should be easier to read both in color and in black & white.

Don't use vectorial files for this type of figures. They contain too many points, and make the pdf file unmanageable!

Our understanding is that Nature Communications requires vectorial figures, so we are afraid there is not much we can do to improve on this point. However, the new figures we have added to replace the previous ones are much lighter and should contribute towards shorter loading times of the pdf file.

Reviewer #2

A relevant portion of the Reviewer's concerns and comments focus on the relationship between the disparity filter and the Pólya filter (DF and PF in the following, respectively). Therefore, before addressing such comments in detail, let us make a few general observations about this broader point. Our goal was to develop a network filtering tool capable of responding to a specific network's heterogeneity and discarding links accordingly. Indeed, our goal was to put forward a novel approach, without any specific intention of establishing the superiority of our filter with respect to preexisting ones (especially since, as we mentioned in the paper, no ground truth exists on what a significant relationship is, which prevents from establishing the superiority of a methodology in a strictly objective manner). To be perfectly honest, the (approximate) equivalence of the DF and the PF for $a = 1$ was not the starting point of our work, but rather a serendipitous discovery. Nevertheless, this relationship links the PF and the DF intimately. Following the Reviewer's comments, we have expanded the characterization of this relationship in both the main text and the Supplementary Information document (as detailed in our answers below). The DF is part of the Pólya family of filters, so the analytical characterization we have developed for the PF automatically transfers to it, enriching its understanding and highlighting its strengths and limitations. On this latter matter, we have added an entirely new section that shows extensive comparisons with the other filters known in the literature. There, we show that the PF family, as a whole, has some optimal features which are not shared by the majority of filters put forward after the DF.

To summarize, we would like to stress that the main strengths of our approach are (*i*) its flexibility, i.e. the possibility of tuning the filter's sensitivity to heterogeneity (including the possibility of using

the heterogeneity of the network under analysis as a null hypothesis for the network itself), and (ii) its analytical tractability, which provides a clear characterization of the features determining whether a link is to be rejected or retained in a certain backbone. To the best of our knowledge, both such properties are entirely new in the network filtering literature, and, most importantly, they allow us to fully characterize and understand the “potential hierarchy of backbones” envisioned in the original DF article.

In the following, we address the Reviewer’s comments in the same order they appear in the report.

The authors did not provide real data examples to demonstrate that the previous method DF will fail because it does not take into account networks’ own heterogeneity. In order to validate the significance of the new method, real data experiments are better included to compare the results of DF and PF. Besides, the authors’ concerns on the over-selection and inflexibility issues of DF are better echoed by real data examples.

We clarified in the above comments that it was not our intention to suggest that the DF “fails” in any way, and we have outlined what are the main advancements introduced by the PF. We have amended the paper accordingly. In addition, we have introduced extensive comparisons between the PF, the DF and other network filtering procedures both in the paper and in the Supplementary Information document. These analyses show that the PF is generally much more aggressive than other filters, regardless the specific value of a , as it usually produces very sparse backbones containing smaller fractions of nodes, links and strength of the original networks. Notably, these backbones are non-trivial for two main reasons: (i) they are made of salient links, which (ii) do not necessarily coincide with the “heaviest” ones. We highlight property (i) by measuring the optimality measure O_1 introduced in the main paper, whereas property (ii) is highlighted by measuring the Jaccard similarity J between the B links that compose a backbone and the heaviest B links in the networks. Values of J close to 1 correspond to fairly trivial backbones, which could have been obtained simply by thresholding.

All in all, our comparisons in the “Comparisons with other network filters” of the paper and in Supplementary Note 8 overall show that the PF backbones are very sparse, salient and non-trivial. These properties are largely shared by the DF, when compared to other available filters. Within this context, the enhancement introduced by the PF is the ability to cover a continuum of backbones, and to select the most appropriate one according to a specific application or metric. In particular, the backbones corresponding to $a = a_{ML}$, i.e. the Pólya urn’s parameter which maximizes the likelihood of observing the specific network under study, are typically ultra-sparse and made of links whose properties are “peculiar” with respect to the network’s own heterogeneity.

In order to show this, following the Reviewer’s suggestion we have added a new section in the paper devoted to two case studies focused on the two networks we already had used in the originally submitted manuscript (which we did in order to avoid “cherry-picking” examples). In the case of the US A airports network, we show how applying the PF induces increasingly high sparsification, as expected. When applying the PF for $a = a_{ML}$, we show that the resulting backbone is only made of a handful of links, which mostly correspond to local short-haul flights. This shows the crucial feature of the PF in action: all the heavy links one would intuitively include in a backbone (such as, e.g., the NY-LA connection and other long-haul flights connecting hubs) are actually discarded by the filter, as they represent fundamental contributions to the network’s heterogeneity. As such, they end up being rejected when tuning the filter’s tolerance to heterogeneity to its very *own* heterogeneity, which is what setting $a = a_{ML}$ accomplishes. Conversely, the links that survive the filtering even under such a high tolerance for heterogeneity are

locally-yet-globally important connections. They mostly correspond to relatively short-distance flights that, however, provide exceedingly important connections to passengers (see, e.g., the number of Alaskan internal flights retained). Such importance is reflected in the global importance of such links in terms of salience (see Supplementary Figure 5).

We also added a case study on the WIOT network, where we have calibrated a macroeconomic model on the full network, and on the backbones obtained with the DF (i.e., the PF for $a = 1$) and the PF for $a = a_{\text{ML}}$. We have used such model to predict the network's weights from year-to-year, and found the highest prediction power to correspond to the latter case. We consider this as further evidence of the PF's ability to identify highly relevant relationships.

The authors argue that the deficiency of DF is partly due to its inclination to select most high-weighted links and its rigidity in the null hypothesis. However, a simple way to address these problems is to adjust the significance level α used in hypothesis testing. For example, the over-selection problem can be solved by decreasing α . In order to make more informative comparisons, the authors should compare PF with DF (with adjustable α 's) and identify the effects that can be achieved by PF only.

We have eliminated a confusing remark in the Introduction of the paper which seemed to suggest that the DF suffers from the drawback mentioned by the Reviewer. This was not our intention, and, indeed, as it can be seen from the comparisons we have added in the revised version, the DF is largely immune from such a drawback.

We thank the Reviewer for suggesting this comparison, which we used as a starting point to demonstrate an insightful relationship between different PF backbones. Let us indicate the set of links included in a given PF backbone as $B(a, \alpha)$, where a is the filter's parameter and α is the statistical threshold used to assess the links' statistical significance. We have added a dedicated part of the Methods Section to show that, given two PF backbones $B(a_1, \alpha_1) \neq B(a_2, \alpha_2)$, one can always find a significance level α_3 that makes the two backbones very similar $B(a_1, \alpha_1) \approx B(a_2, \alpha_3)$. We have demonstrated this result numerically (see Supplementary Note 6) and provided an approximate analytical justification for this (see Methods Section). Now, since the DF can approximately be obtained as a PF with $a = 1$, we can always find a threshold α_{DF} to make the DF backbone very similar to the one extracted by the PF with any pair (a, α_{PF}) . However, even if related, the parameter a cannot be entirely identified with the threshold α because of (i) the analytical control we have over a , and (ii) the fact that α has a precise *a priori* statistical meaning, which is not shared by a .

The point here is that even though the two backbones might even be identical, they would have *completely* different statistical meanings. Indeed, depending on the values of a and α , the same link would be considered validated against a difference tolerance to heterogeneity, and, even more importantly, with a different tolerance for false positives / negatives (encoded by the statistical significance threshold α). In order to exemplify how dramatically different these values can be, in Supplementary Note 6 where we show the values of α_{DF} that must be set for the DF in order to produce the same backbones of the PF for $\alpha_{PF} = 0.05$ and different values of a . As it can be seen, regardless of the multivariate correction in use (Bonferroni or FDR), for several values of a one needs to decrease the threshold α_{DF} in order to allow the DF to produce a backbone very close to the one generated by the PF. In particular, when setting $a = a_{\text{ML}}$ (which, as already mentioned, sets the network's own heterogeneity as the null hypothesis' benchmark) the ratio α_{PF}/α_{DF} is of order 10^{-7} and 10^{-4} when applying the FDR correction on the US Airports and WIOT networks, respectively. Such values indicate that the DF has to be made statistically selective way beyond what is normally considered reasonable in the literature (i.e., values of the univariate threshold α

in the range $10^{-3} - 10^{-2}$) in order to match the backbones produced by the PF for $a > 1$.

In the section “Heterogeneity and weight rescaling”, the authors discuss how the procedure of rescaling all weights with $c \in \mathbb{R}$ affects PF. They demonstrate that the set of selected links vary with respect to c . This property seems to be a disadvantage of the new method, because in real data sets, the absolute scale of link weights could be rather arbitrary. It may not be sensible to let this scale to affect the result of filtering. In contrast, the result of DF is not affected by this scale.

As the Reviewer pointed out, the PF filter in its exact formulation can be applied only to networks with integer weights (e.g., weights coming from a counting process, such as in the US Airports network example). In this context, the PF is able to discriminate between links characterized by the same ratio w/s but by different individual values of w and s (e.g., two links such that $(w, s) = (10, 100)$ and $(w, s) = (100, 1000)$, respectively).

When dealing with networks with non-integer weights, the PF can still be used in the approximate form given by Eq. (6). In this case, the p -value prescribed by the PF is no longer affected by an overall rescaling of the weights. Indeed, Eq. (6) does not assign a p -value to a weight w , but rather to a *rate* of interaction w/s .

In principle, the approximation in Eq. (6) holds when $s \gg k/a$ and $w \gg 1$. Yet, it should be noted that an overall rescaling of the weights by a factor $c \gg 1$ automatically takes us to such a regime. This is relevant to the case of non-integer weights (or, more generally, weights that do not come from a counting process), which are always expressed in terms of a certain scale or unit measure, and, as such, can always be rescaled (see, e.g., the WIOT network, whose weights are expressed in millions of dollars). For example, let us consider a network whose lowest weights are of order 10^{-4} . Applying Eq. (1) of the paper to such a network would entail rescaling its weights by a factor $c \geq 10^4$ before filtering. Doing so would correspond precisely to applying Eq. (6) directly.

We have outlined the same line of reasoning in a new part of the Methods Section, and we have removed the section mentioned by the Reviewer.

The authors argue that one merit of PF is its dependence on r . However, they should notice that DF also has this property. Therefore, this dependence does not necessarily help to show that the new method is better than previous ones.

We never framed the dependence on r as a merit of the PF, but simply as one of its properties. Yet, we believe that such property significantly contributes towards a much deeper understanding of the way the PF works, as it largely determines which links survive the filtering and which do not. As already mentioned, to the best of our knowledge this kind of analytical characterization of the features determining which links are filtered away has never been made available for other filtering methods.

We fully agree with the Reviewer that the above dependence is inherited by the DF, since it approximately is a member of the Pólya family. Yet, we stress that such dependence was not noted in the original DF paper, or in any other paper on the subject, and represents an additional novel contribution of our work.

The section “The backbone family” discusses that by varying a , PF is able to obtain a monotone family of link sets. Similar results can also be achieved by DF with varying α 's. A question similar to that in the second bullet point of the major concerns is, what are the merits of the family obtained by PF compared to that obtained by DF.

We have mostly addressed this issue when replying to the second point raised by the Reviewer, in particular for what concerns the statistical meanings of backbones obtained by tuning the DF's threshold α in order to produce the same backbone generated by the PF for $a > 1$. In addition, we reiterate that the fundamental merit of the PF's family of backbones is the ability to *tune* with respect to the network's heterogeneity. We believe that the aforementioned case studies (particularly the one on the US air transport network) convey this point rather clearly, showing how the PF at high values of a produces ultra-sparse backbones where heavy links and / or connections between hubs are filtered away as a natural component of a network's heterogeneity, and the only links retained are highly salient ones whose structural properties (encoded in the parameter r) are statistically significant even with respect to such heterogeneity.

The authors discuss that the soft dependence of PF on $r = wk/s$ ensures the multiscale nature of the network, while links with small r 's still have a chance to be validated by PF. One question arising from this argument is: how is PF better than the method of doing direct thresholding on r , which seems to be both reasonable and quite simple. More discussions and justifications of PF are needed here.

We thank the Reviewer for bringing this point up, which was not clear enough in our original submission. As mentioned by the Reviewer, the key point is that typically links with a large r are validated by the PF, but links with small r are not necessarily rejected and still have a chance of being retained. The crucial point is therefore that such dependence is soft, so that ultimately each link has to be assessed individually. Thresholding would imply inverting Eq. (7) to determine the value of r_{thr} corresponding to a desired level of significance. However, this would hamper the statistical interpretability of the results obtained, as there would be no well defined null hypothesis underpinning the network backbones, without leading to any major advantage in terms of efficiency, as computing the PF's p -values is not particularly time consuming (see the reply to another point below for more details on this).

Visually, thresholding on r amounts to applying a vertical cut for $r = r_{\text{thr}}$ in the lower panels of Fig. 2, and only retaining the links represented by the points to the right of such a cut. This is essentially equivalent to discarding all the points with $r < r_{\text{thr}}$ lying far away from the dashed lines representing Eq. (7), which only holds when $w/s \approx 0$. Hence, thresholding leads to preferentially discarding links that depart from such a regime. These, in turn, mostly correspond to relatively heavy links attached to low-degree nodes (as low values of k still lead to low values of r when w/s is significantly different from zero).

The above argument illustrates why thresholding reduces the power of the PF in terms of quality of the backbones obtained. As we mention in the paper, one of the main motivations to use the PF is that of incorporating heterogeneity into the null hypothesis against which the statistical significance of links is assessed, including the one of the specific network under study. This possibility is only granted when using the proper PF, and leads to more coherent backbones. In order to convey this point, we have added a new Supplementary Note, where we report the difference in size of the largest connected components obtained by applying the full PF and by thresholding for the corresponding value of r obtained through

the inversion of Eq. (7). As shown the Section, thresholding leads to backbones that are substantially more disconnected than those generated by the full PF.

The PF is presented partly to address DF's over-selection problem of high-weighted links. A natural concern is, will the new method suffer from the problem of under-selection, especially when a is selected by the MLE procedure. Discussions and numerical experiments are better provided to address this concern.

As already mentioned in other responses above, we never meant to imply any problem with the DF in terms of an over-selection of heavy links. However, we realize this was not entirely clear in the original submission. We have changed the paper in order to amend it of any ambiguity in this respect, and we have introduced a number of comparisons (both in the paper and in Supplementary Note 8) to clarify this. Such comparisons make it clear that the DF is already a quite aggressive method when compared against other available options. In this respect, the PF, when applied for values $a > 1$, can undoubtedly be even more aggressive. One of the points we have now made clearer in the paper (which we already have touched upon in previous replies) is that this is not a limitation, but rather a feature to be exploited. Indeed, setting $a = a_{\text{ML}}$ defines the “nullest” hypothesis a network can be compared against within the Pólya family of null hypotheses. To the best of our knowledge, no other filtering methodology provides the opportunity to tailor a null hypothesis around a specific network. This is deliberately meant to be a very restrictive null hypothesis, aimed at identifying links that are “surprising” with respect to the rest of the network. Indeed, if a network’s weights were generated *exactly* by a Pólya process, then the PF for $a = a_{\text{ML}}$ would simply return an empty backbone.

In the second paragraph of the section “The role of the r ratio”, the authors discuss how the outcome of PF varies, when w/s is fixed and k varies. It seems to me that this statement is problematic, since usually when k increases, w/s will decrease. It is natural to decouple the variables s/k or w from k but not w/s from k .

We agree with the Reviewer and thank him/her for having pointed this out. Following the suggestion provided at the end of his/her report, we have now rewritten the definition in Eq. (4) also as $r = w/\langle w \rangle$ in order to highlight its alternative interpretation as relative size between the weight of the link under consideration and the average weight on the links attached to the corresponding node. We have rephrased the paper accordingly, and removed all arguments where we had illustrated the functioning of the PF based on w/s and k separately. We only kept mentions of w/s where strictly necessary, i.e., to justify the approximation in Eq. (7).

The calculation of the p -value of the new method involves the generalized hypergeometric function. As far as I know, the result of a generalized hypergeometric function is not always easy to obtain and I'd like to see more comments about this computational issue here.

The Reviewer is right. Indeed, the numerical computation of the p -values is done through the sum in Eq. (2), where each summand can be computed efficiently and with high precision via Eq. (1). We added a mention to this in the paper, and we have provided a Matlab code to implement the PF (mathworks.com/matlabcentral/fileexchange/69501-polya-filter).

The idea that the section “Heterogeneity and weight scaling” tries to convey is a little tangled and the purpose of this section is not clear.

Following the Reviewer’s suggestion we got rid of this Section, and clarified the points it was trying to convey in two additional subsections placed in the Methods Section.

Reviewer #3

We appreciate the Reviewer’s suggestion to make the paper more accessible. We have done our best to improve the presentation of our method while also getting rid of most mathematical technicalities, which we have moved to the Methods Section and to the Supplementary Information document. In addition, following a suggestion by Reviewer #1, we have introduced a schematic representation of the model in the “The Pólya filter” Section, which we hope will contribute to make our method more intuitive. We believe that the changes we have made in this revised version substantially improved the paper’s readability. In light of these changes, we are even more convinced that our paper will attract significant interest from the multidisciplinary Network Science community, and therefore should be aimed at a broad readership such as Nature Communications’.

In the following, we address the Reviewer’s comments in the same order they appear in the report.

The model only considers the local information of the nodes, how to reflect the influence of network global structure on the significance of links.

The Reviewer is correct in pointing out that our filtering method only relies on local information, but, to the best of our knowledge, this is true for any filtering procedure based on a statistical null hypothesis. In this respect, our suggestions on how to fix the value of the parameter a (see “Fixing the free parameter” Section) were precisely aimed at suggesting strategies to select links locally while still considering their global importance from the viewpoint of the whole network. We have added a sentence in this Section to clarify this point. First, the maximum likelihood procedure allows to devise the value of a corresponding to the Pólya process whose statistics best matches the heterogeneity of the whole network, and is responsive to it (as shown in Supplementary Note 5). Even more to the point, the salience-based procedure to fix a is aimed at maximizing the overall salience retained in the network backbones generated by the PF. We have added a sentence in this sub-section to stress that salience is a non-local property reflecting the network-wide importance of links, e.g., in terms of their contribution to the network’s transport properties.

In the introduction, the authors reviewed many approaches to filtering, which were not used for comparison in the later experiments. Comparisons should be added. Besides the mentioned methods, there are many network-based approaches to find the network skeleton.

We thank the Reviewer for pointing out this limitation of our study. We have added extensive comparisons between our method, two of the methods mentioned in the Introduction of our original submission (now Refs. [25] and [30]), plus three additional methods which we considered for this purpose (Refs. [33], [34], [45]). We conducted such comparisons both on the networks we had already used in our original

submission, as well as in two new network datasets (the High School dataset produced by Ref. [53] and the Florida dry season ecosystem network of Ref. [54]).

As already mentioned in the replies to Reviewer #2, such comparisons have allowed us to make a number of observations. In particular, the PF is generally much more aggressive than other methods, as it leads to sparser backbones. The filter’s aggressiveness obviously depends on the value of the parameter a , and we have stressed this point in the revised version of the paper. In particular, we have stressed how this can be exploited to obtain ultra-sparse backbones by setting $a = a_{\text{ML}}$, where a_{ML} is the optimal value identified via maximum likelihood estimation. This corresponds to the “nullest” hypothesis relative to a network’s *specific* heterogeneity, which in turn typically leads to discarding links that would otherwise be retained by other filtering procedures (e.g., heavy-weighted links connecting hubs) with lower tolerance for heterogeneity. Relationships surviving this filtering are usually “unexpected” links (see for example the case study on the US Airports network we have added, which we also mention in another reply below) whose properties are statistically significant even when assessed against the network’s overall heterogeneity. This is further corroborated by another comparison we performed, using as metrics the Jaccard similarity J between the B links that compose a backbone and the heaviest B links in the networks. Values of J close to 1 correspond to “unsurprising” backbones that could have been attained simply by sorting the links in a network in terms of weight and retaining the heaviest ones. Our analyses show that the PF, for a wide range of values of the parameter a , generates highly non-trivial and salient backbones, whereas other methods often yield backbones that are either “unsurprising” or not salient.

In equation (4), s and k are for original network or for Pólya network? The indicator r is for node or for link? If the latter, which endpoint will be used in equation (4), or there are two r from the viewpoint of the two endpoints. What is the mathematical relation between r and a in equation (4)?

The strength s and degree k we show to in Eq. (4) refer to nodes and links in the original network, before applying the PF, and there is no mathematical relationship between the ratio r and the PF’s parameter a . In the revised version of the paper, this is clarified in the Methods Section, where we show that the r ratio computed from such quantities is the main quantity that explains whether a link will be retained or rejected by the PF. Eq. (4) shows that lower values of r tend to correspond to lower statistical significance in the PF for *any* value of a . However, whether a link is ultimately retained or rejected depends on the specificities of the PF in use, i.e. both on the value of a and on the chosen significance level α .

As clarified at the end of the “The Pólya filter” Section and in Supplementary Note 2, the PF can be used to compute two p -values can be associated to the same link. In the case of undirected networks, these are computed from the perspectives of the two nodes connected by the link under study, i.e. using the strength s and degree k of both such nodes to compute the p -values. In the case of directed links (see Supplementary Note 2), two p -values can still be computed for the same link by considering it an outgoing link of the node it starts from and an incoming link of the node it arrives to, respectively. Regardless of the network’s directionality, we consider a link to be validated by the PF when at least one of its two p -values is below the significance threshold α chosen for the test.

If the equation(6) is more suitable for the network with uniform degree distribution and heterogeneous weight distribution? Heterogeneous degree distribution may lead to a larger error. Therefore, I’m wondering if Pólya filter is a general model that can be applied to all kinds of networks. This should be discussed to show in which condition the model performs well.

The PF is applicable to any weighted network, regardless of its specificities. The same consideration applies to Eq. (6), which holds as soon as the conditions that make it valid apply (i.e., $s \gg k/a$, and $w \gg 1$), regardless of the nature of the underlying network. In Supplementary Note 5 we have included an analysis on synthetic networks aimed at showing under which conditions the PF is more useful. Namely, we showed how the ML estimate for the parameter a , which is essentially a proxy for the heterogeneity of the network, does not change significantly in networks with uniform weights (even when the weight distribution spans several orders of magnitude), and remains well below 1. On the one hand, this shows that the PF, as originally intended, is best suited to filter heterogeneous networks. Nevertheless, this also shows that the PF can be fruitfully applied to filter networks with non-heterogeneous weight distributions by keeping $a \in (0, 1)$.

Why the performances on the two networks shown in figure 2 are much different? How do network structural features matter?

There is no distinctive difference in performance when applying the PF to the two networks shown in Fig. 2. The apparent differences are due to the fact that the WIOT network is significantly more dense than the US Airports network, which in turn increases the density of points around the lines representing Eq. (7). In addition, in the WIOT case the regime under which Eq. (7) is derived (i.e., $w/s \approx 0$) is valid for a larger fraction of links, in line with the “physical” meaning of links in such a network. Indeed, in modern economies complex goods can only be manufactured through the interaction of several industrial partners.

How to evaluate the performance of Polya filter with other benchmarks? One possible method is network dismantling. If a network collapses faster after removing the links according to their significance given by Polya filter than other methods, this to some extent shows the superiority of this method. Other evaluation methods can also be applied.

In order to present comparisons between the PF and other methods, we initially considered network dismantling, as suggested by the Reviewer. However, based on all the papers we found in the literature on the subject, to the best of our knowledge network dismantling is based on the removal of *nodes*. On the other hand, the PF is a link-based procedure, i.e. it assigns a measure of statistical significance to each single link. In this respect, there would be no clear way to implement a network dismantling protocol based on the results of the PF, as one would need to somehow decide whether to remove a node or not based on the information provided by the PF on all its links, and such information can be extremely heterogeneous (i.e., links belonging to the same node can either be highly statistically significant or perfectly compatible with the null hypothesis).

We therefore chose to present a number of other comparisons between the PF and preexisting methodologies, which we have referred to in a previous reply to the Reviewer. In all cases we present comparisons on the following quantities: (i) the fraction of nodes kept in the backbone, (ii) the fraction of edges kept in the backbone, (iii) the optimality measure O_1 introduced in Eq. (5) of the paper, and (iv) the Jaccard similarity J between the B links that compose a backbone and the heaviest B links in the networks. The latter measure is particularly interesting: values of J close to 1 correspond to “unsurprising” backbones that could have been attained simply by sorting the links in a network in terms of weight and retaining the heaviest ones. Our results show that the PF typically produces non-trivial backbones that are very different from those that would be obtained by thresholding on the weights. On the other hand, some of the other methods tend to produce backbones that could be well approximated via thresholding.

The method should be tested on more disparate networks to show its performance. The results can be put in SI.

We have added comparisons between the PF and the methods listed above with two additional datasets (also mentioned above). In addition, we have devoted a section of the main paper to a more detailed study of the application of the PF to the two networks we already included in our original submission.

Is this method only for the weighted network? Can it be also applied to unweighted networks where w values for all links are 1 leading to $r = 1$.

As mentioned in a previous reply, the PF can only be applied to weighted networks, since the main rationale of the test is to assess the statistical significance of repeated interactions between pairs of nodes.

How to apply Polya filter on the networks with non-integer weights, e.g. $w = 0.1, 2.5$?

We thank the Reviewer for bringing up this point, which we have now addressed fully in the Methods Section. As we outline in detail there, the PF can still be applied to networks with non-integer weights through the approximate form given by Eq. (6). In this case, the p -value prescribed by the PF is no longer affected by an overall rescaling. Indeed, when the regime under which Eq. (6) is valid holds, the PF does not merely assess the statistical significance of a weight w , but rather provides a p -value for the observed *rate* of interaction w/s between two nodes, irrespective of the specific weight w .

The paper should be rewritten to improve the readability to make it be of interest to the broad readership of Nat. Comm. For instance, one can add one simple example network (with 6-8 nodes) to illustrate the model and make a comparison with other 2 or 3 benchmarks. Besides, it will be much better if the authors provide some real applications of their method.

Following the Reviewer's suggestion, we have added a simple sketch of the PF functioning (Fig. 1 of the revised manuscript), and wrote a new section devoted to case studies, where we present applications of the filter and show how it unveils interesting information on the US Airports network and the WIOT network. In addition, we have considerably simplified the language in several parts of the paper and removed the most tedious mathematical technicalities from it, in order to make it more accessible to a general audience.

Reviewer #1 (Remarks to the Author):

The authors addressed all my comments. I am in favor of publication.

Reviewer #3 (Remarks to the Author):

There are many dismantling methods by removing links in the literature, such as a recent paper [Complexity 9826243,2018]. One can also find many methods by searching with keywords "edge partition, edge cutting, etc.." For other questions, I'm satisfied with the reply.

Response to the reviews of submission NCOMMS-18-25831A “A parametric approach to information filtering in complex networks: The Pólya filter”

We thank the Reviewers for their comments and feedback, which gave us the opportunity to substantially improve our paper. In the following, we reply to the only comment we have received in the second round of reviews, made by Reviewer 3.

Reviewer #3

There are many dismantling methods by removing links in the literature, such as a recent paper [Complexity 9826243, 2018]. One can also find many methods by searching with keywords “edge partition, edge cutting, etc.” For other questions, I’m satisfied with the reply.

We thank the Reviewer for pointing out the above paper and research stream, which we did not know. We noticed that in most of these works link removal is performed based on link betweenness (or closely related measures). In this respect, we would like to highlight that our method tends to preserve links with a higher salience (as can be seen in Supplementary Figure 5), which is also closely related to betweenness and other measures that are informative of a network’s transport properties. Any link removal protocol based on such measures would produce very similar results. Therefore, a network dismantling protocol based on the Pólya filter would be equally effective, to a first approximation, to other currently available methods.